# Pharmacological Proposal Approach to Managing Chronic Pain Associated with COVID-19

**DOI:** 10.3390/biomedicines11071812

**Published:** 2023-06-24

**Authors:** Grisell Vargas-Schaffer

**Affiliations:** Pain Center, Centre Hospitalier de l’Universitaire de l’Université de Montréal (CHUM), Montréal, QC H2X 3E4, Canada; grisellvargas@gmail.com

**Keywords:** chronic post-COVID pain 2, post-COVID syndrome 3 rehabilitation pain, chronic pain post vaccination, neuropathic pain

## Abstract

Background: Post-COVID syndrome is widespread and chronic pain associated with this syndrome is increasingly being seen in pain clinics. Understanding and managing Chronic Post-COVID Pain (CPCoP) is essential in improving the quality of life of patients. Relevant sections: Identify the types of pain associated with post-COVID syndrome and look for ways to treat them. Results and Discussion: Based on our experience, we have identified five groups within CPCoP: (1) chronic pain post-hospitalization in intensive care or long hospitalizations, (2) pain associated with rehabilitation, (3) exacerbation of existing chronic pain pre-COVID-19 infection, (4) central and peripheral neuropathic pain post-COVID-19 infection, (5) chronic pain post vaccination. To fight against misinformation, we created an information capsule for doctors, nurses, and other health workers at a conference via the ECHO* program, delivered 2–3 times a year. Conclusions: In pandemic and post-pandemic periods, it is important to determine the sequelae that a disease can leave in the general population, and to understand and treat them. The model proposed may serve as an inspiration to other pain centers to treat the increasing number of patients with CPCoP.

## 1. Introduction

Chronic pain associated with COVID-19 is somewhat unknown despite now becoming one of the main reasons for consultation in pain clinics. Understanding and managing Chronic Post-COVID Pain (CPCoP) is essential to improve the quality of life of patients. Patients with chronic, non-cancerous pain often suffer from stigma, and this can be exacerbated when the cause of pain is poorly understood. Thus, doctors and health personnel must be informed to identify and treat this emerging population effectively [1,2,3,4,5,6].

At least 65 million individuals around the world have or have had long COVID [1]. In this review, we used the definition of long COVID proposed by NICE and WHO [2,3]: signs and symptoms that develop during or after an infection consistent with COVID-19, continue for more than 12 weeks and are not explained by an alternative diagnosis. 

The syndrome usually presents with clusters of symptoms, often overlapping, that may fluctuate over time and affect any system in the body [4,5,6,7]. Recent publications note that 1 in 10 patients may have sequelae from COVID-19 [8,9], and only 1 in 4 feel fully recovered at 1-year post-infection [10]. The list of persistent symptoms reported by patients is extensive: chronic cough, shortness of breath, chest tightness, endocrine disorders, gastrointestinal diseases, lung diseases, cardiovascular diseases, chronic kidney diseases, dysautonomia, peripheral vascular diseases, cognitive dysfunction, extreme fatigue, anosmia, parosmia, ageusia, sleep problems, difficulty concentrating, memory problems, depression, anxiety, mood changes, skin lesions, alopecia, and as noted, pain [2,3,5,8]. These symptoms may persist beyond 12 weeks after the primary infection [3].

The syndrome appears to be reminiscent of those seen after other infections [10,11] including brucellosis, Q fever, giardiasis, mononucleosis, Rocky Mountain spotted fever, and viral infections such as dengue, Zika, chikungunya, and others. Even 150 years ago, concepts such as neurasthenia and lethargic encephalitis were noted as fatigue, anxiety, depression, and neuralgia syndromes frequently occurring after infections.

## 2. Chronic Pain Associated with COVID-19

The article centralizes scientific research literature that was conducted by searching some of the most valuable databases: PubMed National Library of Medicine 8600 Rockville Pike Bethesda, MD 20894. Google Scholar United State, Medline, and Embase. MeSH (Medical Subject Headings) United States National Library of Medicine, controlled vocabulary was used for searching in PubMed(United States National Library of Medicine (NLM)), while Emtree (Embase subject headings) University of Manchester Library United State, controlled vocabulary was used for searching in Embase in order to obtain the most associated synonyms for the entered terms. Unfortunately, at present there is a lack of evidence-based recommendations for the pharmacological management of patients with chronic pain associated with COVID-19. This article presents a concise summary of the essential elements that should be used to treat chronic pain associated with COVID-19. A total of 85 bibliographic references are cited to support and validate the information in this paper.

The pain-related manifestations of COVID-19 can be divided into four categories:Painful manifestations associated with vaccines.Pain in the acute phase of the disease and in palliative care.Chronic Post-COVID Pain (CPCoP).The impact of the pandemic on the chronic pain population.

### 2.1. Painful Manifestations Associated with COVID-19 Vaccines

The benefits of vaccination are numerous and clearly outweigh the risks. Vaccines reduce disease, disability, and death from infectious diseases [12,13,14,15,16,17,18].

Vaccines prevent between two and three million deaths each year. With SARS-CoV-2, being unvaccinated increases an individual’s risk of becoming ill and dying from the disease.

The societal and individual risks of being ill and of dying from COVID-19 are greater than the risk of complications caused by the vaccine [12,13,14,15,16,17,18,19,20]. However, a small percentage of patients will have vaccine-related adverse events.

At the time of writing, there are over 440 scientific studies and reports on the adverse effects of COVID-19 vaccines. The WHO reports that cases of anaphylaxis are extremely rare, estimated at between 1/50,000 and 1/1,000,000 patients vaccinated, and are associated with polyethylene glycol (PEG). There are reports of thrombotic manifestations with COVID-19, vaccine including thrombosis and vascular accidents; thrombocytopenia, myalgias and arthralgias, immune alterations, myocarditis and pericarditis, rhabdomyolysis and fasciitis, neurological complications such as facial pain, facial hypoesthesia, Bell’s palsy, Guillain–Barré syndrome, Miller–Fisher syndrome, epilepsies, encephalitis, acute disseminated encephalomyelitis, functional neurological disorders and, unfortunately, death [12,13,14,15,16,17].

Cases of serious adverse events following vaccination against COVID-19 are regularly reported in the scientific and popular literature and may explain the reluctance of some to be vaccinated. For this reason, it is important to reassure patients (see Figure 1), to give them explanations of risk and to treat associated complications if they occur. Thus, healthcare providers need to understand the pathogenesis of different complications, such as nervous system damage, caused by vaccines [17,18,19,20,21,22].

mRNA-based vaccines require encapsulation in lipid nanoparticles (a combination of ionizable cationic lipids, cholesterol, phospholipids, and polyethylene glycol) in order to reach the intracellular machinery to translate the transcripts into proteins [23].

The mRNA modifications used (the mRNA modification m1ψ17) and the components of nanoparticle encapsulation are both potential agents that could explain inflammatory lesions observed in post-vaccination neuropathy. In particular, PEG18 has been implicated as a possible cause of anaphylaxis in patients receiving the Pfizer-BioNtech vaccine (manufactured by Pfizer and BioNtech; is an mRNA-based COVID-19 vaccine developed by the German biotechnology company BioNTech. For its development, BioNTech collaborated with American company Pfizer to carry out clinical trials, logistics, and manufacturing).

The presence of these inflammatory deposits in the endoneurium has been confirmed through nerve biopsies obtained from people who developed neuropathies after vaccination [24].

Serious neurological complications following the administration of a COVID-19 vaccine that can produce chronic pain include acute disseminated encephalomyelitis, cerebral venous sinus thrombosis, transverse myelitis, multiple sclerosis, Bell’s palsy, Guillain–Barré syndrome, Parsonage–Turner syndrome, trigeminal neuralgia, small fiber neuropathy, and brachial plexus neuropathies [22,23,24].

Another set of post-vaccine complications that could explain chronic pain are myositis or rhabdomyolysis. The occurrence of rhabdomyolysis after vaccination is not limited to COVID-19 vaccines. In general, vaccines (specifically adjuvants) can trigger the release of cytokine myotoxic substances, such as tumor necrosis factor alpha, which have been shown to induce skeletal disorders and muscle breakdown. Another potential mechanism may be the generation of cross-reactive antibodies targeting skeletal muscle cells leading to myositis [24,25,26].

### 2.2. Pain in the Acute Phase of the Disease and in Palliative Care

For outpatients with mild and moderate symptoms of COVID-19, the most used pain treatments are acetaminophen and nonsteroidal anti-inflammatory drugs, but there are limited data regarding their use [27,28].

There are currently no clinical trials or specific guidelines regarding the topic of pain management in acute COVID-19 patients [28,29,30]. In intensive care units (ICUs), optimal pain management can be extremely challenging in mechanically ventilated COVID-19 patients who are often deeply sedated and receive neuromuscular blocking medications. In addition to pain symptoms caused by the virus, such as myalgia, arthralgia, sore throat, headache and peripheral neuropathies, problems associated with ICU treatment (procedural pain, prolonged mechanical ventilation, muscle wasting, immobility during prone positioning) may arise. COVID-19, despite its prevalence, is still poorly understood. Thus, for the treatment of acute pain, each patient requires an individual approach based on the available knowledge and the patient’s condition and comorbidities [28,29].

For the palliative care of patients at the end of life due to severe COVID infection, the literature is scarce. Many people are dying from coronavirus, but consensus guidance on palliative care in COVID-19 is lacking [31,32,33,34,35,36]. In addition, there is a lack of availability of palliative care in hospices hit hard by the pandemic and, unfortunately, many deaths have occurred without assistance from palliative care and pain management [35].

### 2.3. Chronic Post-COVID Pain (CPCoP)

Chronic pain associated with long COVID forms part of what is known as Neuro-PACS or Neurological syndrome post-COVID 19, which is characterized by a long list of manifestations and symptoms that affect the central and peripheral neurological system [37]. Table 1 shows the most common neurologic manifestations/symptoms attributed to COVID-19. These can be present with or without pain.

Based on our experience at the CHUM Pain Management Center, we feel that the chronic pain syndrome associated with long COVID can be divided into five groups, as shown in Figure 1. This classification agrees with some published reports [26,38,39,40,41].

Pain present after a procedure or prolonged immobilization is associated with injuries to the shoulders, hips, knees, cervical and lumbosacral spine, strained brachial plexus or peripheral nerves such as ulnar and radial nerves, and meralgia paresthetica. Prolonged hospitalization is also associated with loss of muscle strength and endurance, and degenerative joint damage usually accompanied by joint pain. 

Musculoskeletal pain associated with COVID-19 has a prevalence of 11–50% [38,39,40]. Radiology researchers have noted the appearance of autoimmune myositis occurring with COVID-19. Case reports describe myositis and rhabdomyolysis in patients with long-term COVID, both as a late complication and as a symptom [38,41].

There is hematogenous spread and direct musculoskeletal invasion by SARS-CoV-2 via the ACE2 receptor, which have been proposed as pathophysiological mechanisms for musculoskeletal pain [42,43]. In addition, the inflammatory response with storming and activation of immune cell cytokines and deposition of immune complexes, release of myotoxic cytokines and injury secondary to homolog between human muscle cells and viral antigens have been highlighted as additional mechanisms for pain associated with COVID-19 infection [26,27,42,43].

Central and peripheral neuropathic pain associated with nerve and nervous system damage by SARS-CoV-2 is present in some patients. One hypothesis is that nerve damage is caused by infectious spread via hematogenous and neuronal routes, hypoxia, and immune damage [26,41,42,43]. The penetration of coronavirus to the central nervous system can cause serious neurological damage. 

Vasilevska et al. [43] discuss a possible pathophysiology of anti-NMDAR encephalitis induced by SARS-CoV-2, during an acute infection, in which virus particles including NSP8 and NSP9 are released. The released particles are recognized by T cells, leading to the activation of B cells, which become plasma cells and produce IgM and later IgG anti-bodies against NSP8 and NSP9. The SARS-CoV-2-associated inflammation of the endothelium and IL-17, produced by activated T cells, disrupts the blood–brain barrier, allowing NMDAR antibodies to enter the central nervous system [26,43].

Molecular mimicry also plays an important role in the possible pathophysiology of central and peripheral neuropathic pain in COVID-19. Molecular mimicry can be defined as the structural similarity between foreign molecules (microbial/viral) and mammalian host molecules. The immune system may react inappropriately by trying to damage these cells with similar surface antigens in joints, blood vessels, or in otherwise healthy organs. Currently, molecular mimicry is the dominant hypothesis as to how viral antigens initiate and maintain autoimmune responses that lead to tissue damage [26,42].

Molecular mimicry may explain the injury to peripheral nerves in COVID-19 infection, given the similarities between SARS-CoV-2 surface glycoproteins and glycoconjugates in human nervous tissue. Mimicry of NMDA receptors may contribute to neuropsychiatric symptoms and encephalitis in severe cases of COVID-19 [26,42,43,44,45] and could explain the mechanism of central and peripheral neuropathic pain experienced by patients with long COVID.

The result is a post-SARS-CoV-2 neurological syndrome characterized by different entities, including: headaches, facial necrosis, polyradiculopathy, pain post-AVC, central or peripheral neuropathic pain, Guillain–Barré syndrome, Miller–Fisher syndrome, myopathies, chronic fatigue, post-COVID brain fog (which is characterized by confusion, headache, short-term memory loss), anosmia, parosmia, ageusia, coronasomnia, and various degrees of depression, anxiety, and post-traumatic stress disorder [42,46,47].

Patients with CPCoP present a mix of different types of pain (nociceptive, neuropathic, and nociplastic). Refining the differential pain diagnosis is important to establish an appropriate and personalized treatment plan for patients suffering from CPCoP.

### 2.4. Impact of the Pandemic on the Chronic Pain Population

Several studies carried out during the pandemic demonstrated the negative repercussions of the pandemic on patients suffering from chronic pain [48,49,50]. As reported by Dassieu et al. [49] during the COVID pandemic, four dimensions of chronic pain were described: the first, reinforced vulnerability due to uncertainties regarding pain and its management; the second, the social network as a determinant of pain and psychological condition; the third, increasing systemic inequities intermingling with the chronic pain experience; and the fourth, more viable living conditions due to confinement measures.

Other impacts included a decrease in physical activity, the lack of complementary treatments such as physiotherapy, massages, and acupuncture that were interrupted during the pandemic, increased depression, anxiety, and stress, and fear of being infected when accessing needed healthcare interventions [48,49,50].

## 3. Discussion

In anticipation of a growing number of patients with sequelae of CPCoP, we created a specialized CPCoP unit at the end of 2020 in our pain management center at the University Hospital of Montreal, CHUM. We used a three-step process.

Step one: We began by establishing admission criteria. In order to be able to evaluate and treat as many patients as possible, we required that patients be referred by a physician who would be an active partner in the follow-up of treatment prescribed by physicians in clinic. To be consistent with current practice, we chose the NICE definition of long COVID: Signs and symptoms that develop during or after an infection consistent with COVID-19, continue for more than 12 weeks and are not explained by an alternative diagnosis [2]. 

We included a range of presentations and sequelae, including musculoskeletal, central, and peripheral neuropathic pain, headaches, chest pain, abdominal and vascular pain, as well as chronic pain post-hospitalization, including in ICU. We also included patients who had multiple pain syndromes concurrently and those with post-COVID psychological sequelae such as anxiety, depression, and post-traumatic stress disorder.

Step two: We presented the concept of a chronic post-COVID pain clinic during a general assembly meeting and asked medical staff to collaborate with us. Most referrals arrived via our collaboration with a research project where patients have been screened and received a diagnosis CPCoP.

Step three: We built a flowchart to streamline the assessment and treatment of patients with CPCoP. We favor a multidisciplinary team approach that includes a neurologist, two anesthetists, a family doctor, as well as a physiotherapist. These professionals complete a global assessment and integrate treatment with a personalized exercise program. Additionally, we have formalized a relationship with a rehabilitation center for complex cases that require longer intervention.

The organizational framework for our approach Is presented in Figure 2.

### Treatment of Chronic Pain Associated with COVID-19

It is essential to be clear that the treatment of chronic pain must be multidisciplinary, integrating psychological treatment, physiotherapy, and other disciplines into the pharmacological treatment with an integrative medicine approach. In this way, the patient is maintained as an active participant at the center of the pain management strategy.

The first step in dealing with chronic pain associated with COVID is to explain to the patient that this is a real disease and that it takes time to find the right balance. Globally, in our experience, most patients recover.

The treatment of chronic pain must be based on therapeutic education. Therapeutic patient education is education managed by healthcare providers trained in the instruction of patients. It is designed to enable a patient, or a group of patients and families, to better manage their conditions and prevent avoidable complications while maintaining or improving their quality of life [51,52].

In patients with chronic post-COVID pain, patient education and management should be performed holistically by an interdisciplinary team. This team must give clear guidance to the patient on the skills they need to best manage life with a chronic disease. Then, they must apply existing protocols to treat each type of pain, as in Figure 3.

Figure 3 presents our suggestions for the treatment of CPCoP. Weak or non-conventional opioids (tramadol, tapentadol, and buprenorphine) should be considered before moving to strong opioids [53]. When symptoms of neuropathic pain are present, it is important to use the treatment algorithm for neuropathic pain, where opioids are considered adjuvant medications and not the principal treatment. 

Setting realistic goals with the patient is important. Generally, a 20–30% reduction in pain or reduction of 2 points on the pain scale is achievable and can be considered clinically significant by patients and physicians.

Since most of the chronic pain associated with long-term COVID is neuropathic pain, a neuropathic analgesic ladder should be considered. See Figure 4.

Offer first-step gabapentinoids (pregabalin or gabapentin), tricyclic antidepressants (e.g., amitriptyline and nortriptyline), and serotonin and norepinephrine reuptake inhibitors. Topical preparations are particularly effective in cases of localized allodynia or intolerance to oral co-analgesics. Carbamazepine may be offered first line if the neuropathic pain is trigeminal neuralgia [53,54,55].

The analgesic effect of tricyclic antidepressants appears more rapidly than the antidepressant effect and requires less frequent, small doses. Although, SNRIs appear less effective at reducing pain. They should therefore be offered to patients who present with a depressive state associated with pain, who cannot tolerate tricyclic antidepressants, or in whom the latter are contraindicated.

As a second step, offer other anticonvulsants (topiramate, lamotrigine) and selective serotonin reuptake inhibitors (SSRIs) (duloxetine, venlafaxine).

As a third step, offer other anticonvulsants (levetiracetam, clonazepam) and other drugs such as cannabis, baclofen, clonidine.

Fourth, offer lidocaine or ketamine IV in specialized pain centers. In patients with severe pain, tramadol and tapentadol can be used as co-analgesics at each step. 

Methadone, a strong opioid and full agonist at the μ-opioid receptor and a non-competitive antagonist to the N-methyl-d-aspartate (NMDA) receptor, can be used for severe pain. Start with a dose of 0.5 mg and increase according to the intensity of the pain and observed side effects. If methadone is used to treat CPCoP, follow guidelines on monitoring more severe adverse effects such as QTc prolongation, drug interactions, and hypoglycemia [56].

When the pain is controlled with a relief score improvement of ≥4/10, maintain co-analgesia for at least 24 weeks, then reduce doses gradually thereafter.

Physical/occupational therapy and other complementary techniques can be used at each step to provide tools to patients to reduce pain and functional limitations.

The use of anticonvulsants as neuromodulators should be guided by existing guidelines on their use in neuropathic pain. Anticonvulsants are classified pharmacologically based on their mechanism of action; there is no classification specifically for use in the treatment of neuropathic pain.

Table 2 shows a classification of anticonvulsants used in neuropathic pain with their mechanisms of action, the recommended doses in monotherapy, and the maximum dose. The most used anticonvulsants in the treatment of neuropathic pain are carbamazepine, phenytoin, gabapentin, pregabalin, lamotrigine, clonazepam, topiramate, and levetiracetam.

Table 3 shows the number needed to treat (NNT) for the co-analgesic drugs used in neuropathic pain [57,58,59,60,61,62,63,64,65,66,67,68,69,70,71,72,73,74], which provides an idea of how many patients we need to treat with an anticonvulsant to have a 50% reduction in pain intensity.

Carbamazepine showed the best efficacy (NNT 1.7), while gabapentin and pregabalin had relatively high numbers (NNT 6–8 and 3.8–11.1, respectively), despite being widely used [58,62].

Gabapentinoids have an interesting pharmacological profile and NNT [62] for neuropathic pain. They have been shown to be effective in relieving neuropathic pain in several research models, cause few adverse effects, and do not cause drug interactions [62].

When initiating drug therapy against neuropathic pain, the doctor must advise the patient of the following:The importance of taking the medication regularly and as directed.Dose increases should be made gradually to avoid adverse reactions.The possibility of successive trials if the effect of co-analgesics is unsuccessful.The probability of having to take more than one drug (multimodality).The prolonged use of these molecules in the case of improvement.

Advice when prescribing co-analgesics for the treatment of neuropathic pain: The choice of the molecule depends on the individual patient, possibility of drug interactions, and the side effects of the co-analgesic. It is important to consider the following aspects: Add one co-analgesic at a time to assess the clinical effects. The ideal dose is one that provides significant relief without causing adverse effects. If the chosen co-analgesic is ineffective or intolerable, the molecule should be stopped and replaced by another first-line co-analgesic. In the event that at least one option from each large family of first-step co-analgesics (except in case of contraindications) has been tried without success, consider a second step option; see Figure 4. If a co-analgesic provides partial relief, combine it with an agent with a differing mechanism of action. If the pain is relieved (a relief score improvement of ≥4/10), maintain co-analgesia for at least 24 weeks, then reduce doses gradually thereafter.

Recommendations for stopping co-analgesia are difficult to make; there are no clear rules about weaning. It is suggested to stop one molecule at a time. If pain reappears when stopping a molecule, maintain the minimum dose of that molecule and try to reduce the dose again later (a few months to a year later). If the patient is taking more than one co-analgesic, try to remove a second molecule instead.

We have also found that patients suffering from CPCoP often develop high levels of anxiety. The psychopathological mechanisms of post-COVID-19 depressive and or anxiety symptoms are mainly related to the peripheral immune–inflammatory response triggered by the infection [75,76]. Other reasons for depression and anxiety in infected patients include social quarantine, economic problems, nutrition problems, and stress [77]. Therapeutic education is used also as part of treatment of anxiety to involve the patient in their treatment, as well as to foster improved adherence to the treatment plan and cooperation with the interdisciplinary-nary team [51,52,78,79].

Stigmatization can also affect mental health. The stigmatization of patients suffering from long COVID is multifactorial and associated with its nature as an infectious disease; the ‘fake news’ related to the origin of the disease; vaccines that were highly questioned at the beginning of the pandemic; the diversity of immune responses across the world’s population; the association of long COVID with the mental health and the lack of good public information from health organizations. Unfortunately, patients suffering from chronic pain associated with COVID-19 are currently stigmatized and often do not receive adequate treatment [80,81,82].

To fight against misinformation and the stigmatization of patients suffering from a CPCoP, we created an information capsule dedicated to doctors, nurses, and other health workers at a conference via the ECHO* Douleur program that occurs 2–3 times a year. This presentation has three objectives: know the post-COVID-19 syndrome; identify the different symptoms and types of chronic pain that accompany post-COVID-19 syndrome; establish post-COVID-19 pain treatment guidelines. Whilst education about long COVID may be an important first step, it is not a magic bullet for addressing stigma.

The ECHO* (Extension for Community Healthcare Outcomes) model™ creates knowledge-sharing networks through multipoint video conferencing. These virtual communities of practice are led by interdisciplinary, specialist ‘hubs’, that join with participants (spokespeople) to share knowledge and best practices.

## 4. Future Directions

Even now, we barely understand the pathophysiology of the persistence of symptoms after infection by SARS CoV-2. Various hypotheses exist to explain the persistence of long COVID, including reservoirs of SARS-CoV-2 in tissues, [83,84] immune dysregulation, autoimmunity [1], priming of the immune system from molecular mimicry [41,42,43], microvascular blood clotting with endothelial dysfunction, and dysfunctional signaling in the brainstem and/or vagus nerve [1].

According to the most recent publications [1], 65 million individuals around the world have long COVID. Based on a conservative estimate, this is an incidence of 10% of the more than 651 million documented COVID-19 cases worldwide. The incidence rises to an estimated 10–20% of non-hospitalized cases, 50–70% of hospitalized cases, and around 10% of vaccinated cases [85,86]. These estimates are important, as many countries around the world have no adequate long COVID records.

Chronic pain associated with COVID-19 must be thought of as a non-cancerous chronic pain that must be treated individually, based on pain type. We must apply treatment schemes and guidelines giving a multimodal focus in which psychological and functional re-education are integrated.

To ensure effective treatment and avoid stigmatization, we need to work on several themes: (a) Research in post-COVID infections must continue, to avoid a new global crisis. (b) Training and education must be provided to healthcare providers to be able to diagnose, treat and educate patients. (c) We need a public communications campaign that informs the public about the risks and outcomes of long COVID.

## 5. Conclusions

CPCoP is a real phenomenon, and the number of patients with CPCoP will most likely increase in the near future. Being prepared for this will be of great benefit during both the pandemic and post-pandemic period. Keeping in mind easy schemes to be used in daily clinical practice can help treating physicians to develop personalized treatment schemes based on the type of pain present in patients suffering from chronic pain associated with COVID-19.

Patients appreciate and respond positively to physicians who take the time to listen to them and encourage their self-care efforts. We hope that our approach will benefit others working in this field.

## Figures and Tables

**Figure 1 biomedicines-11-01812-f001:**
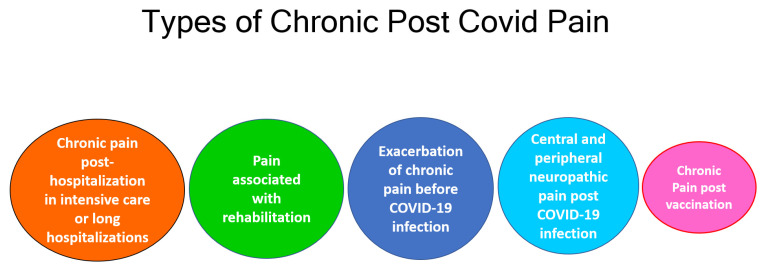
Types of Chronic Post COVID Pain.

**Figure 2 biomedicines-11-01812-f002:**
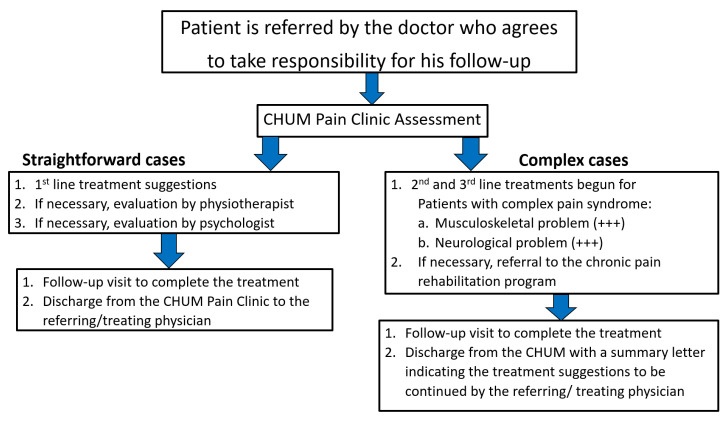
Organizational framework for the approach.

**Figure 3 biomedicines-11-01812-f003:**
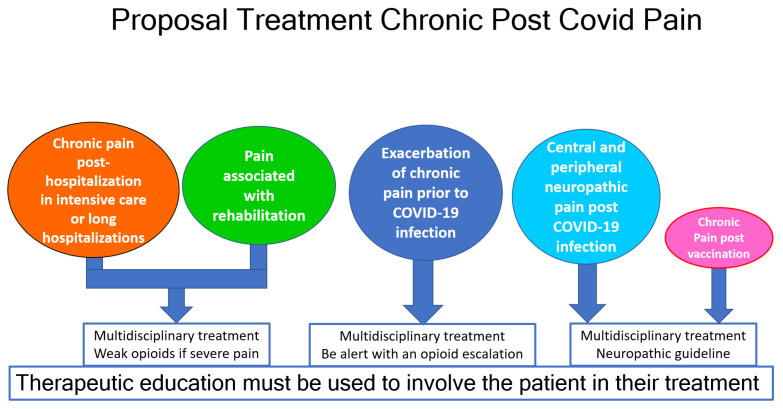
Proposal Treatment Chronic Post COVID Pain.

**Figure 4 biomedicines-11-01812-f004:**
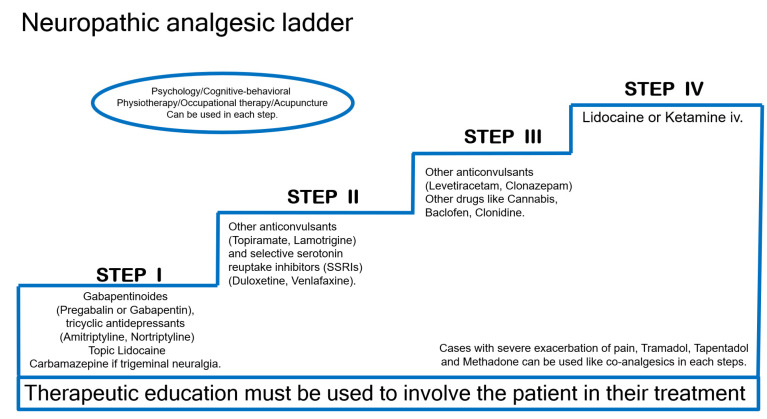
Neuropathic analgesic ladder.

**Table 1 biomedicines-11-01812-t001:** Neurologic manifestations/symptoms attributed to COVID-19.

Alteration of cranial nerves and peripheral nerves.	Craneal nerf dysfunction. Polyradiculitis/Polyneuropathy. Anosmia. Dysgeusia. Tinnitus. Sensorineural hearing loss. Blurred vision.
Movement disorders	Parkinsonism. Status epilepticus. Cerebellar dysfunction. Nonspecific movement disorder. Gait disturbances. Seizure.
Other neurological disorders	Encephalitis/Encephalopathy. Ischemic or hemorrhagic Stroke. Headache. Dysautonomia. Fatigue.
Cognitive and psychological disturbances	Brain fog. Short-term memory deficit. Attention deficit. Depression and/or anxiety. Post-traumatic stress disorder. Obsessive-compulsive symptomatology.

**Table 2 biomedicines-11-01812-t002:** Classification of anticonvulsants used in neuropathic pain.

Anticonvulsant	Classification	Mechanism of Action	Recommended Doses in Monotherapy	Maximum Doses
Gabapentin Pregabalin	Ion channel modulators (reduce neuronal excitability)	Calcium ion channels. Decreasing the density of pre-synaptic voltage-gated calcium channels and subsequent release of excitatory neurotransmitter	300 mg/day 150 mg/day	3600 mg/day 600 mg/day
Topiramate	Multiple mechanisms of action. increasing GABA activity and inhibiting glutamate activity, topiramate blocks neuronal excitability	Canales de iones sódicos receptores GABAA, NMDA AMPA/kainato	100–200 mg/day	1000 mg/day
Carbamazepine Oxcarbazepine Lamotrigine Phenytoin	Ion channel modulators (reduce neuronal excitability)	Sodium ion channels Selectively binds and inhibits voltage-gated sodium channels, stabilizing presynaptic neuronal membranes and inhibiting presynaptic glutamate and aspartate release. Stabilizing the inactive state of the sodium channel and prolonging the neuronal refractory period	100–200 mg/day 600 mg/day 100–200 mg/day 200–400 mg/day	1200 mg/day 2400 mg/day 600–700 mg/day 500–600 mg/day
Levetiracetam	Modulators of the presynaptic junction	Modulation of synaptic neurotransmitter release through binding to the synaptic vesicle protein SV2A in the brain	250 mg/day	1500 mg/day
Clonazepam Diazepam	GABAergic transmission enhancers (potentiate inhibitory neurotransmission)	GABA-A receptors action.	0.5–1 mg/day 5–10 mg/day	2–4 mg/day 15–20 mg/day

GABA, gamma-aminobutyric acid; GABA-A, gamma-aminobutyric acid subunit A. AMPA, α-amino-3-hydroxy-5-methyl-4-isoxazolepropionic acid. NMDA, N-methyl-d-aspartate; Sv2A, synaptic vesicle glycoprotein 2A.

**Table 3 biomedicines-11-01812-t003:** Number needed to treat neuropathic pain.

DRUGS	NNT 50% Decrease in Pain Intensity	Pathologies Where They Are Most Used
Carbamazepine	1.7	Trigeminal neuralgia
Phenytoin	2.1	Refractory NP
Clonazepam	4	Refractory NP
Topiramate	5.29	Headaches
Levetiracetam	4–10	Refractory NP
Gabapentin	6 8	Diabetic neuropathy Post herpetic neuralgia and others NP
Pregabaline	3.8–6.9 8.2 5.9 11.1	Diabetic neuropathy Post herpetic neuralgia and others NP
Lamotrigine	8.3	Central NP
Other drugs used to treat NP		
Baclofen	1.4 (2–2.5)	Atypical trigeminal neuralgia Refractory NP
Ketamine	3–5 >10	Refractory NP Depression
Lidocaine	4–10	Topical application Post herpetic neuralgia. HIV neuropathy Cancer-related neuropathy Refractory NP
Clonidine	4–10	Diabetic neuropathy Post herpetic neuralgia. HIV neuropathy Cancer-related neuropathy Refractory NP
Capsaicin	8.8 (10–12)	Topical application Peripheral neuropathic pain Post herpetic neuralgia. HIV neuropathy Refractory NP
Cannabis	NNTB (30–50) 20–25	Low quality studies
Opioids		
Tramadol and Tapentadol	4.4	NP
Metadona	0	Very limited data for NP

NNT: Number of patients needed to treat a patient reporting a 50% decrease in pain intensity. NNTB: Number needed to treat an additional beneficial outcome (30% pain relief). NP: neuropathic pain in different pathologies.

## Data Availability

Not applicable.

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
