# Peer review of "Pharmacological Proposal Approach to Managing Chronic Pain Associated with COVID-19"

_biomedicines, 2023, doi:10.3390/biomedicines11071812_

Round 1

Reviewer 1 Report

The review addresses a relevant and still topical subject. However, its presentation is quite another story with many inadvertencies detected. The instructions for authors and the template provided by the journal must be read and understood very well. The present review is poorly described in all aspects regarding the management of chronic pain associated with COVID-19. Also, the tables and figures are not relevant in the way they have been presented. Given the space taken up by the graphical elements, 11 pages is too few for a valuable comprehensive review to contribute to the field, especially as it is not the first of its kind. 

No full stop at the end of a title

The abstract sections provided in the template are appropriate and are used for original research articles, not reviews, and should not appear as such.

Nothing has been inserted in keyword 1. Abbreviations are not used for keywords. At the end of the keyword list, there is no period.

L24-29 a whole paragraph without bibliographic references. Please revise.

Introduction- the introduction is poorly organized with little useful information. It should be extensively detailed so as to show 3 distinct areas (without title)- general area, the topic-specific area, and the purpose of the paper, which is the last paragraph of the introduction and should present the contributions and novelty brought to the field, especially as this is not the first time this topic is discussed.

The general management within COVID-19 should be presented in a chapter that can enhance the value of the current article and make it more suitable for Biomedicines. It would be advisable to discuss more the pathophysiological mechanism in order to understand from where the pain may arise, to present the most relevant comorbidities and associated pathologies, some proteins with diagnostic and prognostic roles, and last but not least, the treatment. I suggest checking and referring to the following updated sources: PMID: 36406478; PMID: 35131656; PMID: 36211634.

Chapter 2 should include a methodology chapter presenting the literature search algorithm (Boolean logic operators, filters used, databases used). 

"Relevant sections" is not an appropriate title for a chapter of a scientific article. It needs to be modified and adapted to the content.

L53-L63 no bibliographic reference is inserted for the information mentioned in the text.

Subsection 2.2 is much too short; it would be advisable to expand the data presented based on the latest data in the field.

The graphic elements are very poorly done:

-Table 1 needs to be completely edited - colour, no table head, nothing is understandable from it (organize them according to areas, severity, management, etc.);

-Figure 1 should not appear within the figure itself; no title; no need for abbreviation in the figure as long as it is not used;

-Figures have no title; the flowchart is still the figure.

Figures need to be reorganized in terms of design, clarity, and information because, in their current form, they are not relevant.

L276 table 2 is actually a figure - it needs to be completely redone (the text after it says table 3). Same for the table that is actually the figure in L279

Subsection 2.4. too short (either add more information or link to another subsection).

There are too many bullet point listings. I suggest organizing a part as a text with additional explanations.

The language throughout the text is very informal for a scientific article and should be completely revised (especially in the future directions sections).

Conclusions are too poorly presented in relation to the complexity of the topic addressed.

Author Response

Response to Reviewer 1 Comments

Point 1. Reviewer's comment. The instructions for authors and the template provided by the journal must be read and understood very well.

Response 1: The journal template was followed; the article type structure was respected. You can review the journal article type instructions and see that they were followed.  

Point 2. Reviewer's comment. The present review is poorly described in all aspects regarding the management of chronic pain associated with COVID-19.

Response 2: The management of chronic pain associated with covid-19 is widely described from L218 to L306. It is important to note that so far there are no therapeutic proposals for the treatment of chronic pain associated with long covid.

Point 3Reviewer's comment. The tables and figures are not relevant in the way they have been presented. Given the space taken up by the graphical elements.

Response 3: I believe that the tables are very relevant since they provide an overview of the most used anticonvulsants in the treatment of neuropathic pain and the NNT values integrated in table are too often omitted in everyday practice despite their relevance in therapeutic reasoning.

Point 4. Reviewer's comment. 11 pages is too few for a valuable comprehensive review to contribute to the field, especially as it is not the first of its kind.

Response 4: It is important to note that so far there are no therapeutic proposals for the treatment of chronic pain associated with long covid. Chronic pain treatment already exists and unfortunately there are no new treatments. The article wants to give a guide to remind not only specialists but also general practitioners that post covid infection pain exists and must be treated.

Point 5. Reviewer's comment. No full stop at the end of a title.

Response 5: Corrected.

Point 6. Reviewer’s comment. Nothing has been inserted in keyword 1.

Response 6: keyword 1 is: Chronic Post-Covid Pain.

Point 7. Reviewer's comment. Abbreviations are not used for keywords.

Response 7: Corrected.

Point 8. Reviewer's comment. At the end of the keyword list, there is no period.

Response 8: Corrected.

Point 9. Reviewer's comment. L24-29 a whole paragraph without bibliographic references. Please revise.

Response 9: Corrected.

Point 10. Reviewer's comment. Introduction- the introduction is poorly organized with little useful information. It should be extensively detailed so as to show 3 distinct areas (without title)- general area, the topic-specific area, and the purpose of the paper, which is the last paragraph of the introduction and should present the contributions and novelty brought to the field, especially as this is not the first time this topic is discussed.

Response 10:  Corrected

Unfortunately, at this time there is a lack of evidence-based recommendations for the pharmacological management for patients with chronic pain associated with COVID-19. This article presents a concise summary of the essential elements that should be used to treat chronic pain associated with COVID-19.

Point 11. Reviewer's comment. The general management within COVID-19 should be presented in a chapter that can enhance the value of the current article and make it more suitable for Biomedicines. It would be advisable to discuss more the pathophysiological mechanism in order to understand from where the pain may arise, to present the most relevant comorbidities and associated pathologies, some proteins with diagnostic and prognostic roles, and last but not least, the treatment. I suggest checking and referring to the following updated sources: PMID: 36406478; PMID: 35131656; PMID: 36211634.

Response 11: PMID: 36406478; PMID: 35131656; PMID: 36211634. These articles do not talk about pathophysiological mechanisms of chronic pain in patients with long covid. They talk about comorbidities associated diseases, and risk Assessment in COVID-19; Therapeutic dilemmas; and the potential molecular implications of adiponectin in the evolution of SARS-CoV-2.  

The pathophysiological mechanisms of pain after a COVID-19 infection are explained in lines L133 to L172 and supported by biography (38 to 46).

Point 12. Reviewer's comment. Chapter 2 should include a methodology chapter presenting the literature search algorithm (Boolean logic operators, filters used, databases used).

Response 12: Corrected. Scientific literature research was conducted by searching some of the most valuable databases PubMed, Google Scholar, Medline and Embase. MeSH (Medical Subject Headings) controlled vocabulary was used for searching in PubMed and Emtree (Embase subject headings) controlled vocabulary for searching in Embase in order to obtain the most associated synonyms for the entered terms.

Point 13. Reviewer's comment. "Relevant sections" is not an appropriate title for a chapter of a scientific article. It needs to be modified and adapted to the content.

Response 13: Corrected.

Point 14. Reviewer's comment. L53-L63 no bibliographic reference is inserted for the information mentioned in the text.

Response 14: Corrected.

Point 15. Reviewer's comment. Subsection 2.2 is much too short; it would be advisable to expand the data presented based on the latest data in the field.

Response 15: I agree with your comment unfortunately as mentioned in the text there are no literature/evidence date that support appropriated guideline to manage pain in palliative care in covid infection.

Point 16. Reviewer's comment. Table 1 needs to be completely edited - colour, no table head, nothing is understandable from it (organize them according to areas, severity, management, etc.).

Response 16: Corrected.

The table head is: Neurologic manifestations/symptoms attributed to COVID-19.

Point 17. Reviewer's comment. Figure 1 should not appear within the figure itself; no title; no need for abbreviation in the figure as long as it is not used.

Response 17: Corrected.

Point 18. Reviewer's comment. Figures have no title; the flowchart is still the figure.

Response 18: Every figure has a title. I disagree but Flowchart is not a figure.

Point 19. Reviewer's comment. L276 table 2 is actually a figure - it needs to be completely redone (the text after it says table 3). Same for the table that is actually the figure in L279

Response 19: I disagree with your opinion because Table 2 and 3 are tables, not figures.

Point 20. Reviewer's comment. Subsection 2.4. too short (either add more information or link to another subsection).

Response 20: Corrected.

Point 21. Reviewer's comment. There are too many bullet point listings. I suggest organizing a part as a text with additional explanations.

Response 21: Corrected.

Point 22. Reviewer's comment. The language throughout the text is very informal for a scientific article and should be completely revised (especially in the future directions sections).

Response 22: Prior to submission the article was reviewed and corrected by a medical writer (Nostrum Medical, UK). In my opinion the language is not informal, it is technical language and uses the right language to disseminate relevant scientific information.

Point 23. Reviewer's comment. Conclusions are too poorly presented in relation to the complexity of the topic addressed.

Response 23: Corrected.

Reviewer 2 Report

The manuscript refers to the proposed mechanism to control pain in COVID-19 patients. It is an interesting approach for a pharmacologist; however, the review lacks an adequate association with comorbidities and clinics. Pain control in infectious diseases is not a simple task. Patients with diseases involving decreased mobility and pain can not be treated as patients with other comorbidities that do not suffer chronic pain. Thus, the pharmacological rationale has to start from the condition of the individual and the previous medication. In addition, patients with prolonged COVID-19 differ notoriously, and, in most cases, the events are associated with high production of polyclonal antibodies, which may cross-react with normal proteins causing a temporary effect, ie muscle pain.

The use of scalable therapies to control pain is understandable for general elements, but not in prolonged COVID-19, where a group of individuals may have neurological incapacity limiting therapeutic options.

On the other hand, it is unclear how to approach a descendant scale of medication and what role could exercise and nutrition play in pain resolution.

There are several sentences difficult to understand that should be corrected 

for example chronic pain population, 

Several sentences require revision 

Author Response

Response to Reviewer 2 Comments.

Point 1. Reviewer's comment. The manuscript refers to the proposed mechanism to control pain in COVID-19 patients. It is an interesting approach for a pharmacologist; however, the review lacks an adequate association with comorbidities and clinics.

Response 1: I agree with your comment but, I believe that since it is a review article on the pharmacological aspect, the association with other pathologies is not pertinent.

Point 2. Reviewer's comment. Pain control in infectious diseases is not a simple task. Patients with diseases involving decreased mobility and pain cannot be treated as patients with other comorbidities that do not suffer chronic pain. Thus, the pharmacological rationale has to start from the condition of the individual and the previous medication. In addition, patients with prolonged COVID-19 differ notoriously, and, in most cases, the events are associated with high production of polyclonal antibodies, which may cross-react with normal proteins causing a temporary effect, ie muscle pain.

Response 2: I agree with your comment.

Point 3. Reviewer's comment. The use of scalable therapies to control pain is understandable for general elements, but not in prolonged COVID-19, where a group of individuals may have neurological incapacity limiting therapeutic options.

Response 3: I agree with your comment but for patients suffering from chronic neuropathic pain associated with COVID-19, there is no proposed therapeutic scheme, which is why it is proposed the neuropathic analgesic ladder. 

Point 4. Reviewer's comment. On the other hand, it is unclear how to approach a descendant scale of medication and what role could exercise and nutrition play in pain resolution.

Response 4: I agree with your comment, and I added the follow paragraph: When the pain is controlled with a relief score improvement of ≥4/10, maintain co-analgesia for at least 24 weeks, then reduce doses gradually thereafter.

Reviewer 3 Report

The Author aimed to describe post-COVID pain syndromes.

The topic is interesting but the paper is extremely disorganized. The aim Is actually unsubstantiated. The narrative nature of this review must be acknowledged. Also, the title is misleading. A narrative review can hardly lead to a proposed approach. 

Paragraph 2.2 title is confusing. What does palliative care stands for in this case?

Table 1 can be confounding.

Figure 1 completely un useful

Flowchart 1 is confusing.

Please separate results and discussion sections.

Table 2 unuseful

Conslusions not supported by results

Author Response

Response to Reviewer 3 Comments.

Point 1. Reviewer's comment.  The topic is interesting, but the paper is extremely disorganized.

Response 1: The journal template was followed; the article type structure was respected. You can review the journal article type instructions and see that they were followed. 

Point 2. Reviewer's comment. The narrative nature of this review must be acknowledged.

Response 2:  Prior to submission the article was reviewed and corrected by a medical writer (Nostrum Medical, UK).

Point 3. Reviewer's comment. Also, the title is misleading.

Response 3:  The title was changed.

Point 4. Reviewer's comment. What does palliative care stand for in this case?

Response 4:  Palliative care in the article talks about palliative care in patients with severe COVID-19. For the palliative care of patients at the end of life due to a severe COVID infection, the literature is scarce.

Point 5. Reviewer's comment. Table 1 can be confounding.

Response 5:  Corrected.

Point 6. Reviewer's comment. Please separate results and discussion sections.

Response 6: Corrected. The journal template was followed; the article type structure was respected.

Point 7. Reviewer's comment. Table 2 unuseful

Response 7:  I believe that table 2 is very relevant and provides an overview of the most used anticonvulsants in the treatment of neuropathic, their mechanism of action, the recommended doses in monotherapy and the maximum doses.

Point 8. Reviewer's comment. Conclusions not supported by results.

Response 8:  The article is a review and does not have to support results.

Reviewer 4 Report

Thank you for permitting me to review this manuscript

In this manuscript the authors describe chronic pain and similar entities in association with COVID19

This paper is interesting especially in describing different mechanisms of chronic pain in association with covid 19 

I have minor comments 

Line 42 please provide reference (PPR)

Line 74-75 PPR 

Line 76  I do not see a relation with figure 1 please elaborate 

the division of these entities in  5  subgroups based on experience is not enough , please add a reference or describe somehow your experience data

what are the steps towards the health administration in canada in this context , to my knowledge every thing is coded , are these entities coded?

I see 2 tables 2 ? the second is not proprtionate to others tables 

Author Response

Response to Reviewer 4

Point 1. Reviewer's comment.  In this manuscript the authors describe chronic pain and similar entities in association with COVID19

This paper is interesting especially in describing different mechanisms of chronic pain in association with covid 19

Response 1: Thank you for your comment.

I have minor comments

Point 2. Reviewer's comment.  Line 42 please provide reference (PPR)

Response 2: Corrected

Point 3. Reviewer's comment.  Line 74-75 PPR

Response 3: Corrected

Point 4. Reviewer's comment.  Line 76  I do not see a relation with figure 1 please elaborate

Response 4: Corrected

Point 5. Reviewer's comment.  the division of these entities in 5  subgroups based on experience is not enough , please add a reference or describe somehow your experience data

Response 5: Corrected with references

Point 6. Reviewer's comment.  what are the steps towards the health administration in Canada in this context , to my knowledge everything is coded , are these entities coded?

Response 6: Not everything is coded and in Quebec these entities are not coded

Point 7. Reviewer's comment.  I see 2 tables 2 ? the second is not proprtionate to others tables

Response 7: Corrected

Reviewer 5 Report

Since the chronic pain is associated with psychological conditions, and sleep problems, can the author add the point to the management with antiepileptics for these conditions, and that multi targeting of pain, psychological problems and sleep problems?

Is there any point to be emphasized related to those who have had chronic pain and after covid they have been affected more or needed changes in treatment strategies? Is there any evidence showing that infection with Covid could have been related to worsening of chronic pain patients or Chronic Post-Covid Pain (CPCoP) is an entity that just started in those who were affected by Covid (and have not had any chronic pain before).

Which types of Chronic Post-Covid Pain (CPCoP) more popular? headaches? musculoskeletal pain, others? There are some sporadic publications focused on one type of chronic pain Post-Covid, for example Headache.

Author Response

Response to Reviewer 5

Point 1. Reviewer's comment.  Since chronic pain is associated with psychological conditions, and sleep problems, can the author add the point to the management with antiepileptics for these conditions, and that multi targeting of pain, psychological problems, and sleep problems?

Response 1: Thank you for your comment. When using antiepileptics and anticonvulsants for the treatment of neuropathic pain we can take advantage of the drowsy side effects of these drugs and treat insomnia although it is well known that the best therapy for insomnia is sleep hygiene and cognitive behavioral therapy. I do not think it is necessary to add this to the article since it deals more with pain. If you think it is necessary, I can add this comment.

Point 2. Reviewer's comment.  Is there any point to be emphasized related to those who have had chronic pain and after covid they have been affected more or needed changes in treatment strategies? Is there any evidence showing that infection with Covid could have been related to worsening of chronic pain patients or Chronic Post-Covid Pain (CPCoP) is an entity that just started in those who were affected by Covid (and have not had any chronic pain before).

Response 2: Thank you for your comment.

Unfortunately, up to now there is no published evidence on the exacerbation of pre-existing chronic pain after a Sars-Cov-2 infection. I can share my experience and that is why this group appears in the article in figure 1.

Point 3. Reviewer's comment.  Which types of Chronic Post-Covid Pain (CPCoP) more popular? headaches? musculoskeletal pain, others? There are some sporadic publications focused on one type of chronic pain Post-Covid, for example Headache.

Response 3: Thank you for your comment.

For the prevalence of CPCoP, the literature is scarce. I can share my experience and tell you that neuropathic pain is the most common, followed by musculoskeletal pain and thirdly headaches.

Round 2

Reviewer 1 Report

The present paper presents numerous deficiencies that were not reviewed by the authors even after the first review process, making it unsuitable for publication.

Author Response

Response to Reviewer 1 Comments

Point 1:

The present paper presents numerous deficiencies that were not reviewed by the authors even after the first review process, making it unsuitable for publication.

Response 1: Dear reviewer 1, Thank you for your comments.

In round 1 you made 23 comments and suggestions, all of which were answered and corrected, I sent them 10 days ago. These suggestions and comments enriched the article.

If you have other positive comments to change the article, I am gladly open to consider them and correct the article.

Reviewer 2 Report

The author made an effort to incorporate several points that were raised. As the authors stated is a pharmacologic review mainly for pharmacists. However, it could be useful for academic discussions. 

Only minor mistakes were found in the text

Author Response

Response to Reviewer 2 Comments

Point 1:

The author made an effort to incorporate several points that were raised. As the authors stated is a pharmacologic review mainly for pharmacists. However, it could be useful for academic discussions.

Response 1: Dear reviewer 2, Thank you for your comments.

The manuscrit propose easy schemas to facilitated daily clinical practice where unfortunately clinicians do not have time to consult a pharmacologist to have a neuropathic pain treatment plan for patients.

Reviewer 3 Report

Unfortunately, the Authors did not try to ameliorate his paper according to my previous suggestions, Thus, I still believe that this paper does not merit publication in its current form.

Author Response

Response to Reviewer 3 Comments

Point 1:

Unfortunately, the Authors did not try to ameliorate his paper according to my previous suggestions, Thus, I still believe that this paper does not merit publication in its current form.

Response 1: Dear reviewer 3, Thank you for your comments.

In round 1 you made 8 comments and suggestions, all of which were answered and corrected, I sent them 10 days ago. These suggestions and comments enriched the article.

If you have other positive comments to change the article, I am gladly open to consider them and correct the article.
